# The internal structure of handwriting proficiency in beginning writers

**Lidia Truxius**[1,2]*, **Michelle N. Maurer**[3,4], **Judith Sägesser Wyss**[3], **Claudia M. Roebers**[2]

**1** Institute for Research, Development, and Evaluation, Bern University of Teacher Education, Bern, Switzerland, **2** Institute of Psychology, University of Bern, Bern, Switzerland, **3** Institute for Special Needs Education, Bern University of Teacher Education, Bern, Switzerland, **4** Department of Special Needs Education, University of Oslo, Oslo, Norway

* lidia.truxius@phbern.ch

## Abstract

Fluent and automatized handwriting frees cognitive resources for more complex elements of writing (i.e., spelling or text generation) or even math tasks (i.e., operating) and is therefore a central objective in primary school years. Most previous research has focused on the development of handwriting automaticity across the school years and characteristics of handwriting difficulties in advanced writers. However, the relative and absolute predictive power of the different kinematic aspects for typically developing beginning handwriting remains unclear. The purpose of the present study was to investigate whether and to what extent different kinematic aspects contribute to handwriting proficiency in typically developing beginning handwriters. Further, we investigated whether gender, socioeconomic background, or interindividual differences in executive functions and visuomotor integration contribute to children's acquisition of handwriting. Therefore, 853 first-grade children aged seven copied words on a digitized tablet and completed cognitive performance tasks. We used a confirmatory factor analysis to investigate how predefined kinematic aspects of handwriting, specifically the number of inversions in velocity (NIV), pen stops, pen lifts, and pressure on the paper, are linked to an underlying handwriting factor. NIV, pen stops, and pen lifts showed the highest factor loadings and therefore appear to best explain handwriting proficiency in beginning writers. Handwriting proficiency was superior in girls than boys but, surprisingly, did not differ between children from low versus high socioeconomic backgrounds. Handwriting proficiency was related to working memory but unrelated to inhibition, shifting, and visuomotor integration. Overall, these findings highlight the importance of considering different kinematic aspects in children who have not yet automatized pen movements. Results are also important from an applied perspective, as the early detection of handwriting difficulties has not yet received much research attention, although it is the base for tailoring early interventions for children at risk for handwriting difficulties.

**Data Availability Statement:** All relevant data are within the manuscript and its Supporting Information files.

**Funding:** This work was supported by the Swiss National Science Foundation (SNSF) [grant number

189187]. The funders had no role in study design, data collection and analysis, decision to publish, or preparation of the manuscript.

**Competing interests:** The authors have declared that no competing interests exist.

## Introduction

Even in many nowadays highly digitalized societies, handwriting remains an essential skill and is intensively practiced in the early school years. Handwriting paves the way for numerous other learning opportunities: it enables children to carry out other school-related tasks like math and text writing [1, 2]. The internalization of letters also facilitates reading development [2–4], underscoring its long-term impact on school careers. Imagine Shelley, a first-grade student, copying words (not yet sentences) from the board into her exercise book. Her handwriting is not yet automatized and requires visual, motor, and cognitive control [5–7]. These control mechanisms become apparent when observing Shelley: She adapts her pen movements when task difficulty increases (e.g., longer words, less familiar or more complex letters), and therefore cognitive load increases, she synchronizes her pen lifts, the speed of her copying, the number and length of stops, and her pen pressure [8, 9]. However, it remains unclear how and to what extent these different kinematic aspects contribute to handwriting and whether and to what extent different cognitive processes support Shelley in acquiring fluent and legible handwriting. Therefore, the present study explores (a) the factor structure of early handwriting, considering various kinematic aspects, and (b) investigates how and to what extent handwriting proficiency is linked to individual differences in assumed underlying cognitive processes.

Digitalization also enables investigating handwriting using computerized measures of kinematics (e.g., speed/velocity, stops, pen lifts, pen pressure, and automatization). These kinematic aspects can describe motor aspects of handwriting proficiency relatively comprehensively. One measure often considered to test handwriting proficiency, respectively automaticity or smoothness, is the number of inversions in velocity (NIV) [10–12]. Each pen movement unit is characterized by an acceleration and a deceleration of velocity and, in between, a change (inversion) of velocity. The fewer inversions, the more fluent or automatic handwriting is [13].

The NIV is often used to quantify developmental trajectories and changes in handwriting automaticity. In the beginning, writing movements are strongly controlled and dysfluent. The NIV decreases with increasing practice: writing movements become more automated at age eight and typically reach automaticity around ten years [9, 12]. Wicki et al. [14] investigated the relationship between NIV and higher-order writing in 4th-grade students. The results showed a moderate link between NIV, handwriting speed, and orthographic skills. It thus appears that, overall, the NIV is informative for assessing well-developed and skilled handwriting. The few existing studies including NIV in young and beginning handwriters, point to high NIV, large variations, and strongly dysfluent movements [11, 15, 16]. The NIV thus seems to map the internalizations of motor programs that young children have not yet developed. It is arguable whether the NIV is really the most meaningful indicator of handwriting proficiency for beginning handwriters, as several studies have identified other indicators that could provide valuable insights.

Indeed, comparisons between more proficient and struggling (e.g., dysgraphic) handwriters suggest that other interesting kinematic aspects might be considered to explain early handwriting proficiency. Children with handwriting difficulties in 3rd-grade appear not to differ from proficient writers in speed but in pauses during writing [17, 18]. Notably, there are two types of relevant pauses: pen stops—the pen is immobile on paper, and pen lifts—pen movements in the air to initiate subsequent pen movements. Dysgraphic children differ from typically developing children and adults in both pen lifts and pen stops. In contrast, typically developing children in 3rd-grade do not differ from adults in pen stops but in pen lifts [17]. Furthermore, poor writers typically exert more pressure on paper [19, 20], with high pen pressure potentially leading to cramping and hand fatigue, which in turn might impede smooth pen movements

and impair handwriting development. Together, these findings underscore the potential impact of addressing other kinematic aspects in beginning handwriters, such as pauses (stops and lifts) and pressure control, for a better understanding of handwriting acquisition and handwriting proficiency. In the long run, such an in-depth understanding of early handwriting will enable the detection of early difficulties, known to have vast and negative consequences for a child's school career [1, 2]. However, the relative and absolute predictive power of the different kinematic aspects for typically developing beginning handwriting is still unclear.

Independent of the measure to quantify handwriting being used, there are large individual differences. Two factors potentially explaining handwriting will be tackled in this study: Gender differences and differences as a function of children's family's socioeconomic status. To start with gender, girls usually outperform boys in legibility, spelling, and text quality [16, 21, 22]. The results are somewhat more controversial when considering handwriting processes. While most studies indicate higher writing speed in girls compared to boys [23, 24], findings on other kinematic aspects, particularly NIV, are less consistent. In fourth grade, girls seem to demonstrate more automated sentence copying than boys, but only under time pressure [14]. Among younger children, after one year of handwriting instruction, there are no significant gender differences in the NIV. However, there are distinct associations between fine motor and visuomotor skills with various kinematic aspects. And interestingly, these associations are only observed in girls, suggesting that the underlying processes that contribute to handwriting can differ largely between individuals [16]. At younger ages, typically between four to seven years, girls tend to outperform boys in fine motor tasks [25, 26]. Given the connection between fine motor skills and handwriting proficiency in girls [16], it is possible that girls have an initial advantage when they start handwriting. However, this advantage might diminish as boys catch up in fine motor development over time. We will follow up on this first findings and compare handwriting proficiency between girls and boys at the beginning of handwriting instruction.

Turning to handwriting development as a function of a child's socioeconomic status (SES), children from families of low SES have a higher risk of developing poor literacy skills since low SES households often go along with fewer learning opportunities and poorer learning environments, also including handwriting practice [27]. The impact of SES on reading and literacy development is well established [28]. However, the few existing studies targeting writing suggest that children from lower SES backgrounds tend to have poorer letter knowledge and spelling abilities [29]. Despite the knowledge on less proficient writing abilities, there is a gap in the literature for understanding the specific impact of SES on the acquisition of handwriting. Moreover, children from lower SES backgrounds are also at a higher risk for impaired fine motor development [26, 27, 30]. Although previous studies suggest interindividual differences in writing abilities and fine motor development across different SES levels, it remains unclear whether these effects manifest already early in the development of handwriting and therefore warrant further exploration.

As outlined above, handwriting is a complex multi-component skill requiring fine motor and cognitive control, especially as long as handwriting is not yet automatized [7]. So far, only a few studies addressed which fine motor and cognitive processes are involved in handwriting acquisition. But these studies suggest that executive functions and visuomotor integration, consistently recognized as pivotal in various school-related tasks and indicators for school readiness [31–33] might also affect handwriting [16, 31]. Given their involvement in children's motor development [32–34] and writing performance [35, 36], it is reasonable to assume that they may also play a role in early handwriting.

Executive functions (EF) are understood as a set of basic cognitive abilities supporting concentration, focus on tasks, and self-regulation. EF can be subsumed into three core components: working memory (remembering and maintaining information), inhibition (inhibiting

automated answers), and shifting (switching focus according to circumstances) [37, 38]. While these components are interrelated, they exhibit distinct developmental trajectories [38] and their roles in early handwriting development may therefore vary. In the context of handwriting, a child must simultaneously maintain the written content in mind while retrieving the appropriate letter forms from memory (working memory). The child must also inhibit unnecessary finger movements to ensure precise writing and suppress environmental distractions (inhibition). Additionally, the child needs to shift attention between the writing content and the fine motor finger movements required (shifting). Given these demands, it is not surprising that less well-developed EF skills are typically found to be associated with lower handwriting fluency and writing quality among second graders [39, 40]. In children at the end of first grade, working memory and shifting seem related to different kinematic aspects [16]. However, the relationship between distinct EF subcomponents and handwriting proficiency in children who are just at the beginning of handwriting acquisition, before automaticity becomes established, remains unclear.

As previously mentioned, another important indicator of school-readiness that likely correlates with early handwriting is visuomotor integration (VMI). VMI refers to the ability to process visual information, translate it into a motor answer, and simultaneously monitor the movements. This skill is often tested using a so-called copy design task (with some variations thereof). VMI is likely important for early handwriting because, when children learn to write, they frequently engage in tasks that involve repeated copying of letters and words to internalize the required motor movements. Generally, children who are better able to copy shapes more accurately at ages seven to eight show more legible handwriting, better text quality, and tend to write more fluently [41–43]. Additionally, children with better visual motor memory tend to achieve better writing outcomes supposedly because they can better retain the visual stimuli from their memory and translate them into fine motor movements [44].

The present study focuses on kinematic aspects of pen movements in typically developing beginning writers. So far, there is little knowledge about how different kinematic aspects contribute to handwriting proficiency applying a more broadly defined construct in typically developing beginning writers. Against the background of the existing evidence reported above, where typically only one kinematic aspect had been addressed, we will investigate whether and to what extent different kinematic aspects in pen movements are linked to a broader underlying handwriting factor using confirmatory factor analysis. We expect, besides the NIV, other aspects of handwriting to be relevant for quantifying individual differences in handwriting proficiency. Thereby, we focus on measures that were found to be relevant in more proficient children or in children with handwriting difficulties. Aside from temporal pen movement measures (NIV, pen stops, and pen lifts), we include a measure of force control (press on paper), representing a vital aspect of fine motor control affecting smooth pen movements. Knowledge about the relative importance of these definable kinematic aspects will help to identify the most relevant characteristics of beginning handwriting. This may–in the long run–enable early risk identification and facilitate tailoring interventions accordingly.

Additionally, we considered interindividual characteristics, cognitive, and visuomotor processes that might underlie handwriting acquisition. Given the assumption that girls generally exhibit better fine motor abilities [25], we expected to find gender differences in handwriting proficiency. Additionally, considering recent findings indicating distinct associations between fine motor and visuomotor skills with various kinematic aspects, it is plausible that the development of several kinematic aspects evolves differently between boys and girls. Consequently, we expected variations in handwriting proficiency, which could manifest as differences either in the factor structure or the mean level of the handwriting factor between the two genders. Similarly, regarding SES, we anticipated differences between children from low and high SES

backgrounds. These differences may be found–again–either in the fundamental factor structure or the mean level of handwriting proficiency, influenced by various learning opportunities and fine motor skills among children from different SES backgrounds [26, 27].

Regarding cognitive and visuomotor processes involved in handwriting acquisition, previous research has often reported that EF and VMI are essential in reaching handwriting automaticity [39, 41]. Handwriting binds cognitive resources, which become free when handwriting reaches automaticity. Several studies have focused on the development of handwriting automaticity [11] and assume an important role of cognitive processes for handwriting. However, research investigating the link between cognitive and visuomotor processes in handwriting acquisition is scarce. We hypothesized that more proficient handwriting goes along with better working memory, inhibition, and shifting skills, and superior visuomotor integration.

## Methods

### Participants

We recruited a sample of 1339 first-grade children from public schools in Switzerland. Children with an age more than three standard deviations from the mean age were excluded, as were children with special educational support since we were interested in typically developing children. The final sample consisted of N = 853 children (54.2% girls). Children's age varied between 73 and 96 months ($M$ = 83, $SD$ = 4). 90% of the children were right-handed, 9.7% left-handed, and 0.3% had not yet developed a preference. Most children's first language was German or Swiss German (73%).

We used the International Socio-Economic Index of Occupational Status (ISEI) [45] based on both parents (if available) to estimate children's SES. We divided the ISEI values into quartiles and built four groups to compare children in the lowest SES quartile (14.64 to 44.1; N = 194; 55.2% girls) with children in the highest SES quartile (74.6 to 88.7; N = 193; 52.8% girls).

### Procedure

Parents signed a written consent prior to the study; children themselves agreed verbally to participate on the test days. Trained research assistants tested children in the fall of their first school year. The tests were conducted on two different days individually (handwriting) or in groups of up to seven children (VMI and EF). Every test took approximately 20 minutes.

### Measures

**Visuomotor integration.** To assess visuomotor integration, we used the GRAFOS-2 Screening [46–48]. The GRAFOS-2 Screening is part of the diagnostic instrument GRAFOS-2 developed to assess graphomotor skills from kindergarten to second grade. The GRAFOS-2 Screening is a copy-design task that targets visuomotor integration in a fine-motor context. Similar to other copy design tasks (e.g., Beery-Buktenica Developmental Test of Visual-Motor Integration) [49], children copy different shapes varying in complexity, but they copy each shape six times in small squares ($1cm^2$), which requires finger movements that are important in handwriting acquisition. In the first part of the screening, children copy eight simple shapes. In the second part, they copy five more complex shapes. Eight assistants rated the accuracy of the copied shapes following predefined criteria (0 = incorrect reproduction, 1 = partly correct reproduction, 2 = correct reproduction). Interrater reliability was calculated by comparing

each rater's values with an expert's values for ten children using the weighted Cohens Kappa coefficient. The interrater reliability was good, varying between κ = .66 - .78 [50].

**Inhibition and shifting.** Inhibition and shifting were both measured with the Hearts and Flowers task [51]. The task is often used with children in this age group and is easy to apply in small groups since it is computer-based. The task was administered on a laptop computer with two external response buttons connected to the computer, which are placed on the right and left sides of the screen, and record accuracy and response time (with milliseconds' accuracy). Audio instructions were given via headphones. The task includes three blocks of trials: a congruent block (for establishing a prepotent response), an incongruent block (measuring inhibition), and a mixed block (measuring shifting). In the congruent block, a heart appears on either side of the screen, and children are instructed to press the button on the same side as the heart occurs (24 trials). In the incongruent block, a flower appears on either side of the screen, and children are instructed to press the button on the *opposite* side to where the flower occurs (36 trials). In the mixed block, the two preceding rules are combined (60 trials). Children practiced each block prior to the task started.

For the analyses reported below, reaction times shorter than 200 milliseconds (ms) were excluded as they are typically considered anticipatory responses (1.1% of all trials). Additionally, reaction times that exceeded three standard deviations of an individual's mean reaction time were excluded (1.9%). Blocks with an accuracy lower than 50% were omitted (2.7% incongruent, 1.4% mixed). For the incongruent block (representing inhibition) and the mixed block (representing shifting) we calculated two separate integration scores. The integration scores consider both accuracy and reaction time since the participants were instructed to answer as accurately and quickly as possible. We used the Rate Correct Score which represents the number of correct answers per second [52] that is an adequate score to evaluate EF in school children [53, 54].

**Working memory.** To assess working memory, we used a Backward Color Span task [55]. Task instructions were given via headphones, and children answered on a laptop computer with a touchscreen. The Backward Color Span task is embedded in a cover story about a dwarf losing colored discs. Sequences of differently colored discs appear on the screen, and children are instructed to remember and select the colors in reverse order from a palette of six colors. Prior to the task, children undergo four practice trials. After practice, the task begins with six sequences of two discs each (first block). The sequence length was increased by one additional disc if a child had recalled at least three trials correctly. The task was terminated after four incorrectly recalled sequences within a block. We used the number of correctly recalled sequences for the analyses reported below.

**Handwriting.** For the handwriting task, children copied four different German words (manuscript style) of the same length (six letters) with one word presented at a time. The words were selected from teaching materials to ensure they were age appropriate. Children copied the words on a piece of paper in light grey bars (height of 1 cm) to control for writing size. They wrote the words with a special inking pen on the paper placed on a digitized tablet (WACOM Intuos PRO). This technology, together with the software CSWin PRO 2016 [56] allows quantifying the different kinematics during writing, that is, the temporal and spatial measures of pen movements during writing. The accuracy of spatial resolution was 0.1mm (x-axis and y-axis), and the recording frequency was 200 Hz. Using inductive measurement methods, the pen movements can be registered when the pen is lifted from the tablet (max. 1 cm). The software uses non-parametric regression methods and kernel estimates to calculate accelerations and velocities of pen movements [57]. This study focused on the following kinematic variables: NIV, pen stops, pen lifts, and pen pressure since these represent frequently used and theoretically grounded kinematic variables in dysgraphic and experienced children. We

considered the number of inversions in velocity (NIV) as a measure of automaticity or fluency. The NIV represents the average number of changes in velocity of a movement unit (stroke). The more automatized handwriting, the smaller the number of velocity changes [13]. For pauses, we considered two different measures: pen stops and pen lifts. Pen stops are measured in milliseconds and start when the pen is immobile ($> 200$ms) on paper and end when it continues. Pen lifts are also measured in milliseconds and are represented by the time when the pen is lifted from the paper. Pen pressure on the writing surface is determined by strength (N) whereas one unit represents 101,97 grams. Besides the kinematic pen movements, we asked teachers to rate a child's current global achievement in handwriting classes on a 5-point Likert scale (1 = below average; 5 = above average).

## Data analyses

Values exceeding three standard deviations from the sample's mean were defined as outliers. The number of exclusions in the different tasks and measures varied between one case (working memory) and 23 cases (pen pressure).

We conducted a confirmatory factor analysis (CFA) to test a one-factor handwriting proficiency model using four z-transformed writing variables (NIV, pen stops, pen lifts, and pen pressure on paper). The z-transformation was calculated using pooled mean and standard deviations for the entire sample. We entered one kinematic measure for each theoretically relevant handwriting characteristic to estimate handwriting proficiency balanced and to prevent one single theory-driven characteristic from too strongly dominating the estimated handwriting proficiency latent factor. Therefore, the following measures were finally included: NIV, pen stops, pen lifts (time-variant), and pen pressure (force control). Data for the CFA was analyzed using MPlus 8 [58]. We evaluated model fit with the Chi-square test, comparative fit index (CFI), the root mean square error of approximation (RMSEA), and the standardized root mean square (SRMR). A good fit is indicated by CFI values higher than .95, RMSEA values lower than .06, and SRMR values lower than .08 [59]. Hu and Bentler [59] recommend focusing on CFI and SRMR indexes for large samples. In a second step, we tested the one-factor handwriting solution in different groups (boys *versus* girls and low *versus* high SES) using a multigroup CFA. We tested measurement invariance by comparing different levels of invariances using the Chi-square difference test: 1) configural invariance to test equivalency of the number of factors and pattern of factor loadings across the groups; 2) metric invariance to test equivalency of regression coefficients relating to the latent variable (factor loadings); 3) scalar invariance to test equivalency of the intercepts across the groups. Thereby, a non-significant Chi-square difference indicates invariance.

After conducting a CFA, to address individual differences, we calculated a weighted sum score [60] of writing and correlated the score with VMI and EF. We used RStudio [61] for these analyses.

## Results

Table 1 shows the mean and standard deviations for each writing variable in the entire sample and each subgroup.

First, a confirmatory factor analysis (CFA) was conducted to test the one-factor handwriting structure for the total sample. Fig 1 provides the latent model of handwriting, considering the different kinematic aspects. Overall, the model provided a good model fit ($\chi^2 = 11.40$, df = 2, $p = .003$, RMSEA = .076, CFI = .99; SRMR = .02). All standardized factors loaded significantly on the general handwriting proficiency construct. However, pen pressure showed low factor loadings, whereas NIV and pen stops showed high factor loadings ($\lambda = .80$ and $\lambda = .92$).

**Table 1. Descriptives of handwriting variables separated by groups.**

| | Full sample | Gender | | SES | |
|---|---|---|---|---|---|
| | | girls | boys | low | high |
| | M (SD) | M (SD) | M (SD) | M (SD) | M (SD) |
| NIV | 8.97 (3.48) | 8.52 (3.35) | 9.48 (3.55) | 9.01 (3.52) | 8.63 (3.29) |
| Pen stops (ms) | 1859 (1510) | 1602 (1328) | 2162 (1651) | 1870 (1436) | 1739 (1434) |
| Pen lifts (ms) | 13406 (4655) | 12562 (4254) | 14389 (4907) | 13387 (4450) | 12775 (4609) |
| Pen pressure | 1.41 (.47) | 1.37 (.44) | 1.45 (.51) | 1.39 (.47) | 1.43 (.47) |
| Teacher's rating of handwriting achievement | 3.25 (.99) | 3.41 (.93) | 3.05 (1.02) | 3.07 (.98) | 3.50 (1.01) |

*Note*. NIV = Number of inversions in velocity.

Fig 2 shows the latent factor model of handwriting proficiency for boys *versus* girls. The model showed a good model fit for boys ($\chi^2$ = 7.28; df = 2; $p$ = .026; RMSEA = .08; CFI = .99; SRMR = .02), as well as for girls ($\chi^2$ = 4.42; df = 2; $p$ = .11; RMSEA = .05; CFI = .99; SRMR = .02). In both models, all variables loaded significantly on the underlying handwriting proficiency factor. The tests of invariance (see Table 2) show good fit indices for all models. Constraining the factor loadings and intercepts did not result in a worse model fit based on Chi-

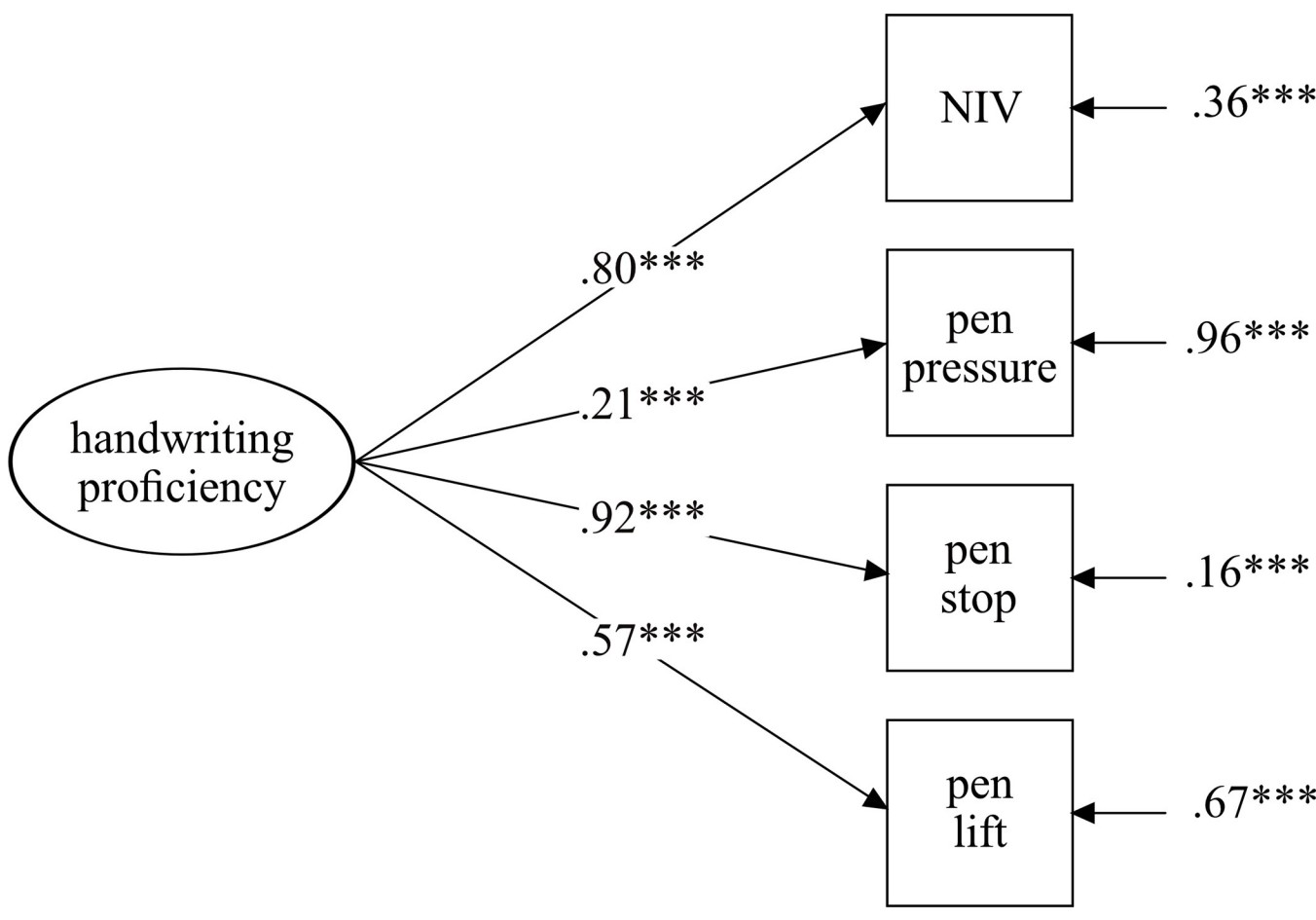

**Fig 1. One-factor handwriting model.** Values represent factor loadings and error terms.

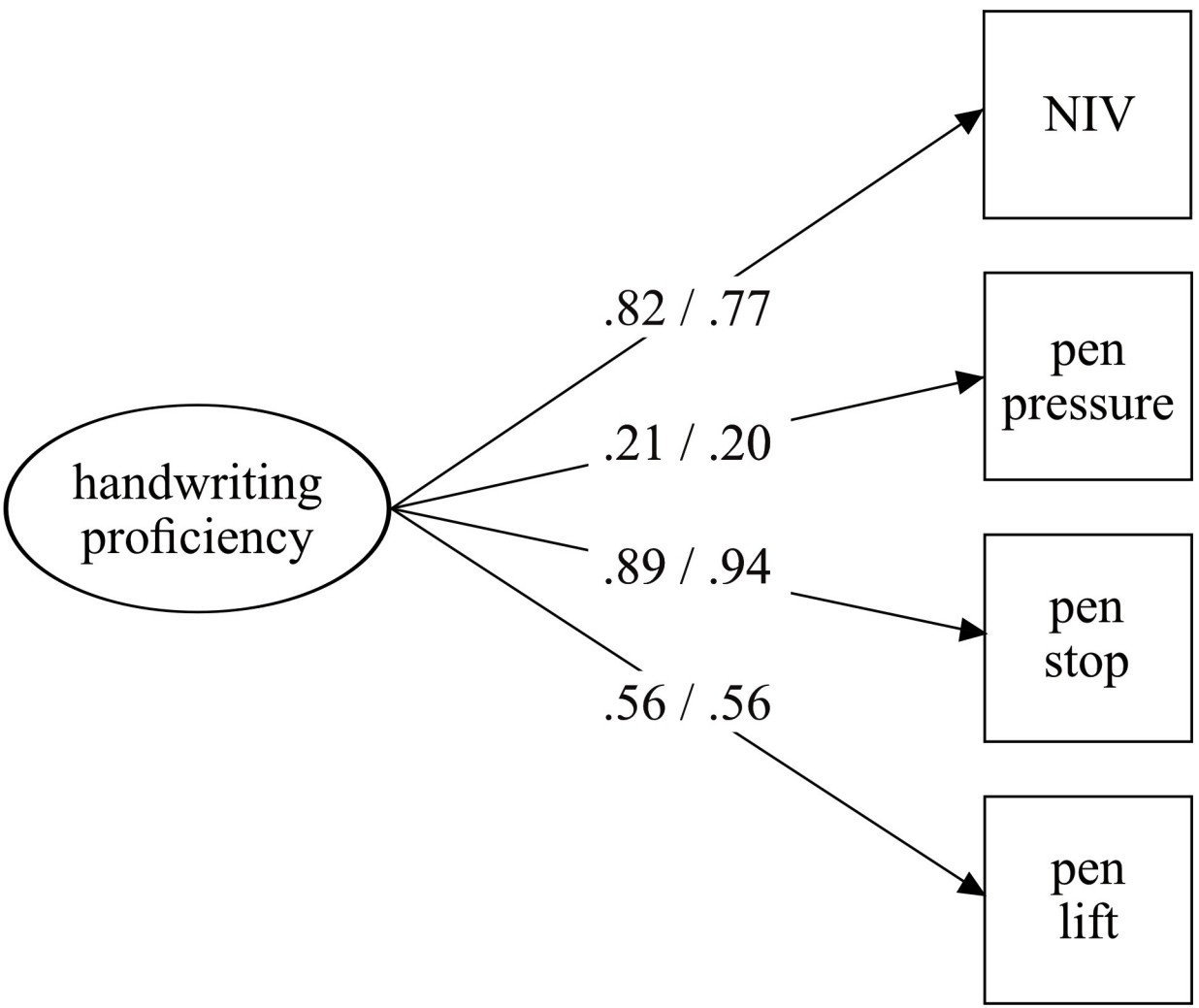

**Fig 2. Multigroup CFA for boys and girls.** Boys before slash and girls after slash. All factor loadings are significant ($p < .001$).

square differences (i.e., metric and scalar invariance). Therefore, measurement equivalence is given; that is, both girls' and boys' handwriting proficiency is best captured with the same factor structure, thus allowing group means comparison. Regarding mean differences in handwriting proficiency, girls showed significantly more sophisticated handwriting than boys ($\Delta M$ = -.35, $SE$ = .06, $p < .001$).

Fig 3 shows the latent factor model for low *versus* high SES children. The model showed good model fit for low SES children ($\chi^2$ = 1.11; df = 2; $p$ = .57; RMSEA = .00; CFI = 1.00; SRMR = .02), as well as high SES children ($\chi^2$ = 2.51; df = 2; $p$ = .28; RMSEA = .04; CFI = .99;

**Table 2. Invariance testing.**

|  | $\chi^2$ | df | $p$ | RMSEA | CFI | SRMR | $\Delta\chi^2$(df), $p$ |
|---|---|---|---|---|---|---|---|
| Boys and girls |  |  |  |  |  |  |  |
| configural invariance | 36.57 | 4 | < .001 | .14 | .99 | .01 |  |
| metric invariance | 37.93 | 7 | < .001 | .10 | .99 | .02 | 1.36(3), $p$ = .72 |
| scalar invariance | 42.06 | 10 | < .001 | .09 | .99 | .02 | 4.12(3), $p$ = .25 |

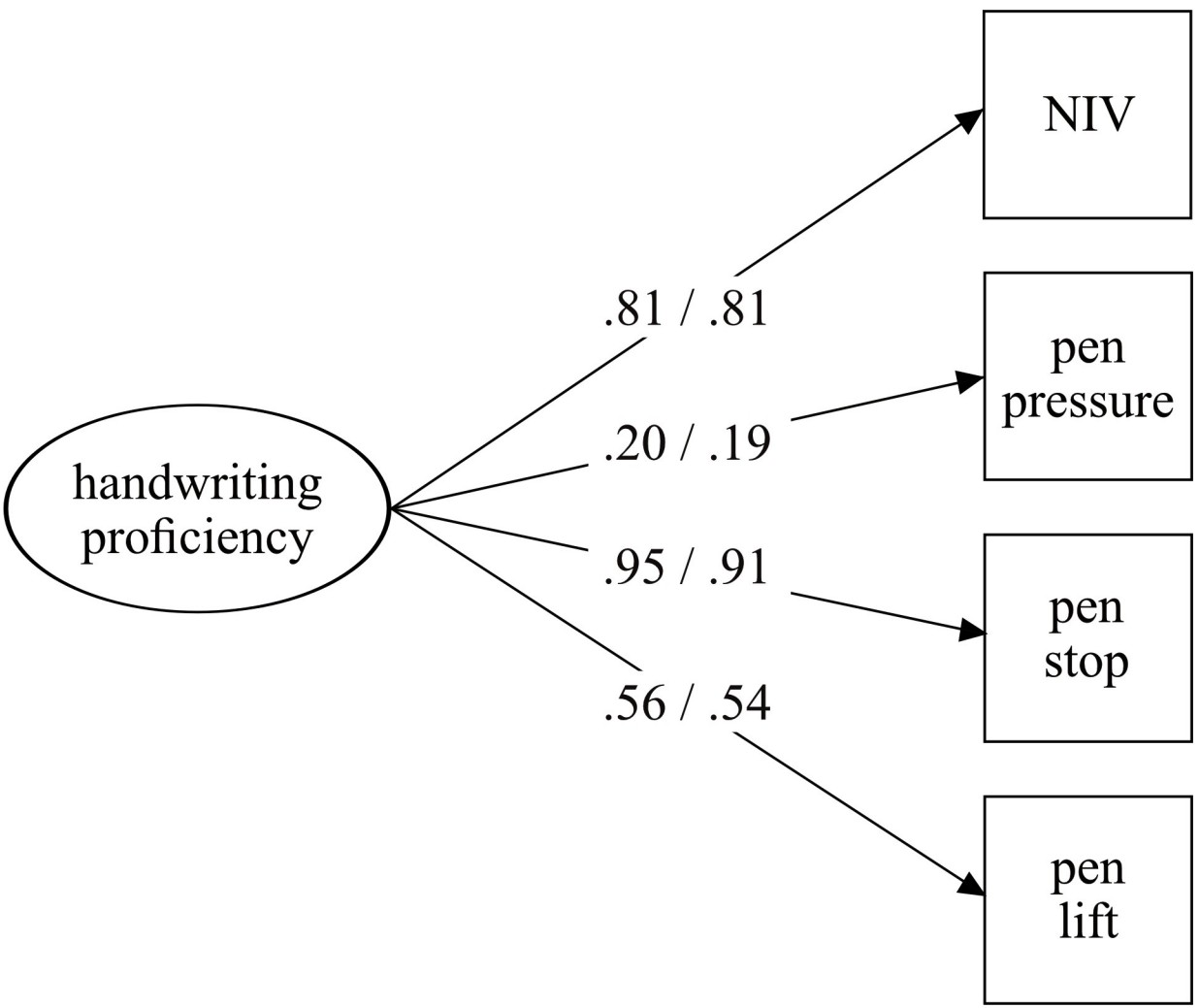

**Fig 3. Multigroup CFA for low and high SES.** Low SES before slash and high SES after slash. All factor loadings are significant ($p < .001$).

SRMR = .02). In this model as well, all variables loaded significantly on the underlying factor. The tests of invariances are reported in Table 3. All models provided a good model fit of the data, and—contrary to our hypotheses—all Chi-square differences were non-significant. Since measurement invariance was given, we compared the group means between low and high SES children. There were no significant differences in handwriting proficiency between the low and high SES groups ($\Delta M$ = -.13, $SE$ = .11, $p$ = .244).

In the next step, we investigated individual differences. To investigate the relation between handwriting, VMI, and EF, we first calculated a weighted sum score representing an individual

**Table 3. Invariance testing.**

|  | $\chi^2$ | df | $p$ | RMSEA | CFI | SRMR | $\Delta\chi^2$(df), $p$ |
|---|---|---|---|---|---|---|---|
| SES |  |  |  |  |  |  |  |
| configural invariance | 22.80 | 4 | .001 | .16 | .98 | .02 |  |
| metric invariance | 23.20 | 7 | .002 | .11 | .98 | .03 | .40(3), $p$ = .94 |
| scalar invariance | 25.89 | 10 | .004 | .09 | .99 | .03 | 2.69(3), $p$ = .44 |

**Table 4. Correlations for handwriting, VMI and EF.**

| | 1 | 2 | 3 | 4 | 5 | 6 |
|---|---|---|---|---|---|---|
| 1. One-factor writing | - | | | | | |
| 2. VMI | -.03 | - | | | | |
| 3. Inhibition | -.01 | .04 | - | | | |
| 4. Shifting | -.03 | .01 | .64*** | - | | |
| 5. Working Memory | -.11** | .24*** | .17*** | .17*** | - | |
| 6. Teacher's rating of handwriting achievement | -.11** | .36*** | .11** | .09** | .24*** | - |

Note. One-factor handwriting, according to the model proposed previously

*$p < .05$

** $p < .01$

*** $p < .001$

handwriting score according to our postulated handwriting model (see above). Specifically, we multiplied our standardized kinematic variables from the CFA by their factor loadings and added the values to a sum score for each child separately. Then, the sum score was correlated with VMI and EF measures and the teacher's rating (Table 4). There was a small correlation between writing and working memory ($r = -.11$), whereas—against our hypotheses—the correlations with VMI, inhibition, and shifting were all non-significant. Furthermore, there was a significant correlation between the one-factor handwriting variable and the teacher's rating of handwriting achievement ($r = -.11$).

## Discussion

The present study aimed to contribute to a better understanding of handwriting proficiency in typically developing beginning writers. Previous studies have often focused on the NIV as a measure of automaticity. Since young children have not yet reached automaticity in handwriting, we included multiple kinematic aspects that, from a theoretical perspective, had proven to be relevant for early handwriting development (pen stops, pen lifts, and pressure). As our confirmatory factor analyses showed, all of these different kinematic aspects contributed to handwriting proficiency, but not all to the same extent. Pen stops contributed most substantially to the underlying construct of kinematics of handwriting. Additionally, we showed that handwriting proficiency in beginning writers is affected by gender and circumscribed cognitive processes.

In line with the literature on more proficient handwriters we found that even in young children who have not yet automatized handwriting, the NIV can meaningfully explain proficiency. Nevertheless, pauses should also be considered as characteristics of handwriting proficiency, whereas our results indicate that the number of pen stops is more indicative of handwriting proficiency than the number of pen lifts. These findings are consistent with previous studies showing longer pen stops in dysgraphic children than in typically developing children, but no differences in pen lifts between the two groups [17]. Pen stops seem to disrupt pen movements, affecting handwriting fluency and smoothness. Pen lifts, in contrast, might be considered an aspect of continuous pen movements and fluency. The initiation of the upcoming letter element affords lifting the pen and placing it at the right location to continue tracing a line on paper. As a consequence, pen lifts might be an essential part of writing and might therefore have less impact on handwriting proficiency per se. Furthermore, as shown in studies of dysgraphic children [19], children's handwriting proficiency is dependent on pen pressure,

even though our findings indicate a small to negligible effect. Pen pressure is a measure of force control, whereas NIV, pen stops, and pen lifts measure the pen's movements. Since all the other variables were time-based measures, including characteristics of movement speed, it is not surprising that there were only small factor loadings for pen pressure. Nevertheless, the small effect of pen pressure on handwriting suggests that force control still slightly affects how the child guides the pen on the paper. Future studies should consider several aspects of grip force and their relevance to handwriting proficiency since grip force seems relevant for handwriting legibility [62] and is unrelated to pen pressure [63].

Results from the multigroup CFA revealed that similar kinematic aspects are involved in handwriting acquisition for boys and girls. In both groups, and similar to the whole sample, pen stops and NIV best explain differences in handwriting proficiency. However, a notable difference emerged concerning the level of handwriting proficiency between boys and girls, with girls outperforming boys. These findings align with our expectations, considering that girls generally develop fine motor skills earlier, potentially granting them an advantage in the initial stages of handwriting acquisition [25].

Contrary to our hypothesis, we did not find different factor structures or mean level differences between children from families of higher versus lower SES. Although the SES differences in our sample were not as pronounced as SES differences in other countries (USA; countries in South America), we had expected structural or mean-level differences as children with high SES are more likely to engage in early practice and typically have more learning opportunities. We conducted a post-hoc test to explore potential mean level differences in our different handwriting variables. The post-hoc test revealed no mean differences for the separate kinematic aspects between children from different SES backgrounds. However, significant differences were observed in teacher's rating of handwriting achievement, with children from higher SES backgrounds receiving significantly higher ratings than those from lower SES backgrounds ($F(1,385) = 18.14$, $p < .001$). This finding might suggest that socioeconomic differences might be more noticeable in observable handwriting outcomes (i.e., legibility, spelling), rather than in the way children form their letters, as measured by kinematic aspects. Or another explanation might be that teacher's ratings were unconsciously but substantially influenced by teacher's knowledge of family SES differences and their implicit theories [64]. Taken together, our findings suggest that children from lower SES backgrounds start their handwriting acquisition at least at a comparable level regarding pen movement control compared to higher SES peers. Importantly, although our findings might hold for beginning handwriting, the impact of SES differences might become stronger when writing becomes more complex, that is, when it concerns spelling and the production of texts. With increasing linguistic elements involved in writing, SES and the home literacy environment might become more crucial [27].

Turning from interindividual differences in the underlying construct to cognitive processes explaining individual differences in early handwriting, working memory appeared to be a key cognitive process for developing handwriting in the present study. Handwriting proficiency in beginning writers was associated especially with working memory but not inhibition or shifting. This suggests that initial handwriting tasks place demands on the naturally limited working memory capacity, especially when handwriting is not yet automatized. For children with lower working memory capacity, handwriting can pose a significant challenge consuming a substantial portion of their limited cognitive resources. In contrast, children with higher working memory capacity may manage various aspects of handwriting simultaneously and with greater ease. Although the observed relationship was modest, these findings highlight the relevance of working memory not only for higher-order writing but also for the basic motor aspects of writing. Therefore, our findings support the assumption that cognitive processes are involved in not-yet-automatized handwriting. Contrary to previous studies in dysgraphic

children, however, we did not find a link between handwriting proficiency and inhibition or shifting [31, 39, 65]. Probably, these executive functions become more important in more sophisticated writing, such as text generation, requiring more planning abilities.

Contrary to our expectations, VMI was unrelated to handwriting proficiency in beginning writers. We expected that children with better VMI skills would be more proficient in handwriting, as they would have enhanced abilities to translate visual stimuli into motor movements, resulting in smoother handwriting. However, it is essential to consider that the task we used in this study was a copy-design task, which requires children to accurately reproduce symbols. This task might have a stronger association with neatness and legibility of handwriting rather than handwriting processes (i.e., kinematics). It is important to note that both VMI and legibility ratings, despite having defined evaluation criteria, possess a subjective element whereas kinematic aspects are more objectively measured. However, previous studies have shown that VMI measures are not only predictive for handwriting legibility but also for overall writing quality [41, 42].

Our findings also contradict previous studies involving older children with handwriting difficulties which have reported impairments in both VMI and kinematic aspects [18, 31]. Since children with handwriting difficulties often show lower VMI and working memory capacity simultaneously [43], the impairment of handwriting processes in dysgraphic children might be an expression of lower working memory capacity. Another explanation might be that typically developing children rely more on working memory than VMI during writing. Since they have well-developed VMI, they might have more cognitive capacity to process visual information and, therefore, process writing elements as units in working memory. Despite the large error terms of the four kinematic aspects in the writing model, the model's fit indices suggest a good model fit. The error terms indicate that further aspects are involved in handwriting, which were not included in this study. Nevertheless, our models provide a first insight into the kinematic aspects that might be considered in typically developing beginning writers. Further, although we did not find any or only minor effects of EF and VMI on handwriting proficiency, this study is one of few studies targeting early handwriting proficiency among beginning writers. Task difficulty could explain why we found small to negligible effects for cognitive aspects and differences between the SES groups. In our study, children copied simple words that were not very demanding. The associations might become more relevant when task difficulty increases, for example, because spelling skills become increasingly involved [66]. Future research should consider the different effects of EF, VMI, and interindividual differences on handwriting kinematics and higher-order writing that contains linguistic knowledge.

Taken together, our study highlights the importance of considering different kinematic aspects when investigating typically developing beginning handwriters. In future approaches, the NIV should not be regarded as the only measure of handwriting proficiency. As this study revealed, other kinematic aspects, such as pauses (e.g., pen stops and pen lifts), explain handwriting proficiency to a comparable extent. Teachers should provide learning opportunities to practice writing movements and internalizing letterforms to support handwriting development. These practice opportunities might be especially important for poor handwriters and children with low working memory capacity as their writing movements are less mature.

## Supporting information

**S1 Dataset. Dataset underlying the findings of this study.**
(CSV)

## Author Contributions

**Conceptualization:** Lidia Truxius, Michelle N. Maurer, Claudia M. Roebers.

**Data curation:** Lidia Truxius.

**Formal analysis:** Lidia Truxius.

**Funding acquisition:** Judith Sägesser Wyss.

**Investigation:** Lidia Truxius, Judith Sägesser Wyss.

**Methodology:** Lidia Truxius, Claudia M. Roebers.

**Project administration:** Judith Sägesser Wyss.

**Resources:** Michelle N. Maurer, Judith Sägesser Wyss.

**Software:** Michelle N. Maurer.

**Supervision:** Michelle N. Maurer, Claudia M. Roebers.

**Writing – original draft:** Lidia Truxius, Claudia M. Roebers.

**Writing – review & editing:** Lidia Truxius, Michelle N. Maurer, Judith Sägesser Wyss, Claudia M. Roebers.

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
