## [Decision Letter · Decision Letter 0]

20 Oct 2023

PONE-D-23-22716The internal structure of handwriting proficiency in beginning writersPLOS ONE

Dear Dr. Truxius,

Thank you for submitting your manuscript to PLOS ONE. After careful consideration, we feel that it has merit but does not fully meet PLOS ONE’s publication criteria as it currently stands. Therefore, we invite you to submit a revised version of the manuscript that addresses the points raised during the review process.

We look forward to receiving your revised manuscript.

Kind regards,

Patrick Charland

Academic Editor

PLOS ONE

Reviewers' comments:

Reviewer's Responses to Questions

**Comments to the Author**

1. Is the manuscript technically sound, and do the data support the conclusions?

Reviewer #1: Yes

Reviewer #2: Yes

2. Has the statistical analysis been performed appropriately and rigorously? 

Reviewer #1: Yes

Reviewer #2: Yes

3. Have the authors made all data underlying the findings in their manuscript fully available?

Reviewer #1: Yes

Reviewer #2: No

4. Is the manuscript presented in an intelligible fashion and written in standard English?

Reviewer #1: Yes

Reviewer #2: Yes

5. Review Comments to the Author

Reviewer #1: I enjoyed reading this paper, and think this topic is an important one that is clearly relevant to the readers of the journal. I have some suggestions below which I hope the authors will find useful, and that I think would improve the manuscript.

Line 1: when the authors say ‘more complex tasks’, do they mean, ‘more complex elements of a task’? In other words, if the motor elements of the task are automatized and fluent (e.g., they can control the pen more effectively), there is more resource to focus on the other aspects of the task, e.g., the narrative or ideas within a composition task, or the operations within a maths task. I realise it’s a nuanced difference, but I think it’s more precise.

Line 2/3: Has most research focused on advanced writers? There does seem to be a reasonable amount on younger children in the existing literature?

Line 81: When the authors stated that it is ‘arguable whether NIV is the most meaningful’, I wondered why they were making that argument. They do then go on to justify this in the next paragraph, but I think it would help the flow to say it directly after the statement.

Line 120/21: I didn’t understand this sentence? Are the authors saying that SES affects handwriting proficiency, but previous research hasn’t looked at this relationship in younger children? If so, is that an accurate reflection of the existing literature?

Paragraph 123-134: I think the authors need to expand a little on why EF is important for handwriting, rather than say it’s obvious. I sympathise that when stating what EF comprises, it does seem relatively obvious, but just a little expansion on why needing to hold onto information whilst attending to a higher goal, or why inhibiting information, is so important. They could do this via a concrete example. For example, a child has to hold on to the goal of writing about what they did over the weekend, whilst also holding in mind grammar rules, etc etc.

Also, do previous studies investigating handwriting look at all aspects of EF in relation to handwriting? Do they use a composite EF score, or look at components separately? Some studies looking at the structure of EF in childhood have shown that EF in young children tends to fit a one-factor or two-factor model (i.e., not necessarily the tripartite model with the separate components of WM, Inhibition, and task-switching).

Paragraph 135-139: This paragraph feels a bit ‘stuck on’ and I think the authors need to expand a little on why VMI is important to study in relation to handwriting. The authors might also be interested in a paper by Waterman et al (The ontogeny of visual motor memory and its importance in handwriting and reading: A developing construct. Proceedings of the Royal Society: B, 282) which looks at how a visual WM task is associated with children’s writing.

Line 141: I would take out the sentence, “Research examining distinguishable aspects…” I don’t think it adds anything and the following sentences are more informative.

Line 158: I would not use the phrase “fine motor processes” to refer to the VMI task. The handwriting task itself requires fine motor processes. And VMI is a visuomotor skill task. So I think ‘fine motor processes’ should be replaced with ‘visuomotor skill’ or ‘visuomotor processes’.

Line 164/165: Why might SES impact the underlying components? I understand why they might predict children from low SES groups might have worse handwriting. But why might we expect, e.g., pen lifts to differ in children from low vs high SES groups in how they contribute to a model of handwriting?

Line 179: What was an extreme age deviation?

Line 220: This section read as if inhibition and task switching were a combined measure. But they appear to be added in to the model separately. Can this be clarified?

Line 235: I didn’t really understand this sentence?

Line 406-408: I did not understand this sentence. Did the authors look at associations between teacher’s ratings and SES?

Line 409-414: This is all too speculative. Unless the authors measured opportunities for writing/pen skills at home for children in this sample, they cannot say that the lack of an association between SES and handwriting suggests that children from low SES families *are* getting opportunities at home.

Line 419: Need to be careful re: implied causality. This study does not allow us to conclude that high WM leads to improved handwriting. So there is a need to be careful re: language used, and acknowledge that they can’t be sure about causal direction. I did feel that the authors also needed to expand on why we might see this relationship between WM and handwriting (whilst being mindful of causality!). For example, they could discuss in terms of limited cognitive resources (because WM is limited). Handwriting, for beginners, places a heavier demand on these limited resources – more effort is required for processes that are non-automatized. But where children have higher WM capacity, this is less problematic. Thinking about the underlying mechanisms will help tie this back into the theory better.

Line 426: the same is true for VMI. Why might we expect VMI to be associated? What did previous studies say about this relationship? I would also suggest the authors should consider some of the negatives of VMI – it is subjectively measured on a scale of 0-2. So, there are clear issues with the test, I would suggest!

Reviewer #2: Clear and coherent proposal: it constitutes an interesting contribution to the field.

The hypotheses that are tested should be presented more clearly before the methodology section (which, moreover, seems very clearly presented to us).

Furthermore, it appears necessary to better define the role of executive functions and visuomotor integration functions in the development of hand writing: as it stands, their role is too quickly outlined. It is the same for the inhibition and shifting measures, for which it is difficult to understand how they are taken into account in hand writing proficiencies.

6. PLOS authors have the option to publish the peer review history of their article (what does this mean?). If published, this will include your full peer review and any attached files.

Reviewer #1: **Yes: **Amanda Waterman

Reviewer #2: No

---

## [Author Response · Author response to Decision Letter 0]

22 Nov 2023

Reviewer #1: I enjoyed reading this paper, and think this topic is an important one that is clearly relevant to the readers of the journal. I have some suggestions below which I hope the authors will find useful, and that I think would improve the manuscript.

Line 1: when the authors say ‘more complex tasks’, do they mean, ‘more complex elements of a task’? In other words, if the motor elements of the task are automatized and fluent (e.g., they can control the pen more effectively), there is more resource to focus on the other aspects of the task, e.g., the narrative or ideas within a composition task, or the operations within a maths task. I realise it’s a nuanced difference, but I think it’s more precise.

Answer: Thank you for this comment. Your wording is indeed more precise, and we were happy to adapt the sentence accordingly. 

Line 2/3: Has most research focused on advanced writers? There does seem to be a reasonable amount on younger children in the existing literature?

Answer: We agree that this phrasing was a bit too unspecified. While there is indeed research available on early writers, when it comes to studying kinematic aspects of handwriting, existing studies fall into two categories: There are either studies investigating the development of handwriting across the primary school years (Rueckriegel et al., 2014) or studies investigating characteristics of handwriting difficulties in older/more advanced writers (Paz-Villàgran et al., 2014, Rosenblum et al. 2003). Notably, there are only very few studies focusing on handwriting kinematics in beginning writers (i.e., Fitjar et al. 2021). 

We have revised the wording to be more accurate in the Abstract. Further, we now try to better highlight this research gap in the introduction. 

Moreover, since our initial submission, a new article was published that targets a similar age group and strongly related to our research questions (Maurer, 2023). We updated the introduction and discussion with regard to these findings. 

Line 81: When the authors stated that it is ‘arguable whether NIV is the most meaningful’, I wondered why they were making that argument. They do then go on to justify this in the next paragraph, but I think it would help the flow to say it directly after the statement.

Answer: Thanks for this comment. We changed the wording and now move to the next paragraph more directly. 

Line 120/21: I didn’t understand this sentence? Are the authors saying that SES affects handwriting proficiency, but previous research hasn’t looked at this relationship in younger children? If so, is that an accurate reflection of the existing literature?

Answer: We recognize that the sentence was unclear. To provide more context, we have included information about existing literature on writing achievement in young children of low SES. Additionally, we have revised the statement about the existing research gap for better clarity. 

Paragraph 123-134: I think the authors need to expand a little on why EF is important for handwriting, rather than say it’s obvious. I sympathise that when stating what EF comprises, it does seem relatively obvious, but just a little expansion on why needing to hold onto information whilst attending to a higher goal, or why inhibiting information, is so important. They could do this via a concrete example. For example, a child has to hold on to the goal of writing about what they did over the weekend, whilst also holding in mind grammar rules, etc etc.

Also, do previous studies investigating handwriting look at all aspects of EF in relation to handwriting? Do they use a composite EF score, or look at components separately? Some studies looking at the structure of EF in childhood have shown that EF in young children tends to fit a one-factor or two-factor model (i.e., not necessarily the tripartite model with the separate components of WM, Inhibition, and task-switching).

Answer: Thank you for this recommendation. We were happy to add a few explanations. The relationship between EF and handwriting should be clearer now. 

Previous studies have focused on a unitary EF factor. Although the multiple-component structure of EF in this age group is controversial, there are still distinguishable developmental trajectories that indicate separable constructs and that warrant addressing the components separately. Furthermore, we were interested in the differential pattern of relationships between EF components and handwriting processes as not all subcomponents might relate equally to handwriting in beginning writers. 

Paragraph 135-139: This paragraph feels a bit ‘stuck on’ and I think the authors need to expand a little on why VMI is important to study in relation to handwriting. The authors might also be interested in a paper by Waterman et al (The ontogeny of visual motor memory and its importance in handwriting and reading: A developing construct. Proceedings of the Royal Society: B, 282) which looks at how a visual WM task is associated with children’s writing.

Answer: Thank you for this constructive comment. We provided more information on the relevance of VMI on handwriting. Furthermore, we are glad to add the reference of the recommended paper. Indeed, it provides another perspective on how VMI and handwriting could be related. 

Line 141: I would take out the sentence, “Research examining distinguishable aspects…” I don’t think it adds anything and the following sentences are more informative.

Answer: Thank you for this comment. We deleted this sentence. 

Line 158: I would not use the phrase “fine motor processes” to refer to the VMI task. The handwriting task itself requires fine motor processes. And VMI is a visuomotor skill task. So I think ‘fine motor processes’ should be replaced with ‘visuomotor skill’ or ‘visuomotor processes’.

Answer: We agree that it is more precise to use visuomotor processes instead of fine motor processes. To avoid any confusion, we replaced fine motor processes with visuomotor processes consistent throughout the manuscript. 

Line 164/165: Why might SES impact the underlying components? I understand why they might predict children from low SES groups might have worse handwriting. But why might we expect, e.g., pen lifts to differ in children from low vs high SES groups in how they contribute to a model of handwriting?

Answer: We realized that we created a misunderstanding in our approach. From a methodological perspective, it is not possible to have differences in structure and mean level at the same time since it is not possible to test the mean when the structure differs. 

Additionally, we agree that little evidence supports the hypothesis of factor structure over the hypothesis of mean level. For these two reasons, we reformulated the hypothesis to be more open and tried to explain that either of these differences is possible. 

Furthermore, a more recent study addressing gender differences led us to reformulate the hypothesis about gender differences. The findings that fine motor and visuomotor skills are differently related to handwriting kinematics in boys and girls might also become evident in factor structure. 

Let us explain here that given that children from low SES backgrounds are assumed to have fewer opportunities to practice handwriting and precursors thereof, the strategy they might be using while writing and drawing can differ. For example, the way they hold the pen might differ (the grip changes during development and experience), altering the pressure that is measurable and changing the fluency of movements (not all grips are ideal for handwriting but may be unproblematic for coloring shapes). These differences might show in the kinematic aspects we measured, too, and that’s why we addressed differences in the factorial structure. 

Line 179: What was an extreme age deviation?

Answer: Children's ages exceeding more than three standard deviations of the mean age were excluded. We added this information to the manuscript. 

Line 220: This section read as if inhibition and task switching were a combined measure. But they appear to be added in to the model separately. Can this be clarified?

Answer: Thank you for paying attention to this small but important detail. We rephrased this paragraph to make it clearer. 

Line 235: I didn’t really understand this sentence?

Answer: We acknowledge that this sentence was not clear. The wording was changed. 

Line 406-408: I did not understand this sentence. Did the authors look at associations between teacher’s ratings and SES?

Answer: We acknowledge that this sentence might be confusing. We did post-hoc tests on the mean differences between the groups (Descriptives are reported in Table 1). It is indeed a little bit confusing since we did not report the results of the post-hoc tests. We added the results of the post-hoc tests and elaborated the findings more explicitly.

Line 409-414: This is all too speculative. Unless the authors measured opportunities for writing/pen skills at home for children in this sample, they cannot say that the lack of an association between SES and handwriting suggests that children from low SES families *are* getting opportunities at home.

Answer: We admit that this interpretation was too speculative and therefore re-worded this aspect. 

Line 419: Need to be careful re: implied causality. This study does not allow us to conclude that high WM leads to improved handwriting. So there is a need to be careful re: language used, and acknowledge that they can’t be sure about causal direction. I did feel that the authors also needed to expand on why we might see this relationship between WM and handwriting (whilst being mindful of causality!). For example, they could discuss in terms of limited cognitive resources (because WM is limited). Handwriting, for beginners, places a heavier demand on these limited resources – more effort is required for processes that are non-automatized. But where children have higher WM capacity, this is less problematic. Thinking about the underlying mechanisms will help tie this back into the theory better.

Answer: Thank you for this very important comment and suggestion for a better explanation. We were glad to adjust the interpretation accordingly. 

Line 426: the same is true for VMI. Why might we expect VMI to be associated? What did previous studies say about this relationship? I would also suggest the authors should consider some of the negatives of VMI – it is subjectively measured on a scale of 0-2. So, there are clear issues with the test, I would suggest!

Answer: Yes, this is a reasonable point. Thank you for bringing this up. Nevertheless, we believe that the measure of VMI is valid, as many items were rated, the screening instrument is commercially available and provides information about reliability and validity. But, yes, in the revised version we now discuss the VMI more thoroughly and mention possible limitations of this instrument. 

 

Reviewer #2

Reviewer #2: Clear and coherent proposal: it constitutes an interesting contribution to the field.

The hypotheses that are tested should be presented more clearly before the methodology section (which, moreover, seems very clearly presented to us).

Answer: In the revised version of the manuscript, you will find the hypotheses clearly and explicitly stated in the last paragraphs before the method section. 

Furthermore, it appears necessary to better define the role of executive functions and visuomotor integration functions in the development of hand writing: as it stands, their role is too quickly outlined. It is the same for the inhibition and shifting measures, for which it is difficult to understand how they are taken into account in hand writing proficiencies.

Answer: Thank you for this important comment. As you will see, we adapted the paragraphs. We tried to give examples and describe the assumed underlying processes of these variables for kinematic aspects of handwriting. 

Reviewer 2 remarks that data are not fully available. We uploaded the data file with the initial submission and carefully reviewed it during revision. However, we are uncertain about the specifics of what might be missing or insufficient. Please, let us know how the data file can be improved.

---

## [Editor Report · Decision Letter 1]

6 Dec 2023

The internal structure of handwriting proficiency in beginning writers

PONE-D-23-22716R1

Dear Dr. Truxius,

We’re pleased to inform you that your manuscript has been judged scientifically suitable for publication and will be formally accepted for publication once it meets all outstanding technical requirements.

Kind regards,

Patrick Charland

Academic Editor

PLOS ONE
---

## [Editor Report · Acceptance letter]

28 Dec 2023

PONE-D-23-22716R1 

PLOS ONE

Dear Dr. Truxius, 

I'm pleased to inform you that your manuscript has been deemed suitable for publication in PLOS ONE. Congratulations! Your manuscript is now being handed over to our production team.

Kind regards, 

on behalf of

Dr. Patrick Charland 

Academic Editor

PLOS ONE